# Measuring Temperature in Coral Reef Environments: Experience, Lessons, and Results from Palau

**Patrick L. Colin [1],\* and T. M. Shaun Johnston [2]** 

[1]  Coral Reef Research Foundation, P.O. Box 1765, Koror 96940, Palau
[2]  Scripps Institution of Oceanography, University of California, San Diego, La Jolla, CA 92093, USA; tmsjohnston@ucsd.edu
\*  Correspondence: crrfpalau@gmail.com

**Abstract:** Sea surface temperature, determined remotely by satellite (SSST), measures only the thin "skin" of the ocean but is widely used to quantify the thermal regimes on coral reefs across the globe. In situ measurements of temperature complements global satellite sea surface temperature with more accurate measurements at specific locations/depths on reefs and more detailed data. In 1999, an in situ temperature-monitoring network was started in the Republic of Palau after the 1998 coral bleaching event. Over two decades the network has grown to 70+ stations and 150+ instruments covering a 700 km wide geographic swath of the western Pacific dominated by multiple oceanic currents. The specific instruments used, depths, sampling intervals, precision, and accuracy are considered with two goals: to provide comprehensive general coverage to inform global considerations of temperature patterns/changes and to document the thermal dynamics of many specific habitats found within a highly diverse tropical marine location. Short-term in situ temperature monitoring may not capture broad patterns, particularly with regard to El Niño/La Niña cycles that produce extreme differences. Sampling over two decades has documented large T signals often invisible to SSST from (1) internal waves on time scales of minutes to hours, (2) El Niño on time scales of weeks to years, and (3) decadal-scale trends of +0.2 °C per decade. Network data have been used to create a regression model with SSST and sea surface height (SSH) capable of predicting depth-varying thermal stress. The large temporal, horizontal, and vertical variability noted by the network has further implications for thermal stress on the reef. There is a dearth of definitive thermal information for most coral reef habitats, which undermines the ability to interpret biological events from the most basic physical perspective.

**Keywords:** water temperature; coral reefs; internal waves; upwelling; advection; climate change; coral bleaching

## 1. Introduction

Water temperature is the most easily quantified physical parameter for shallow-water tropical marine environments. Satellite-derived sea surface temperature (SSST) is used by coral reef scientists for global perspectives on temperature (hereafter abbreviated as "T") patterns/trends and coral bleaching status/predictions [1–5]. There is an "implicit assumption" [6] regarding SSST data that "surface temperature metrics provide useful environmental information with respect to corals that typically live meters to tens of meters below the surface". Satellite data have been thoroughly accepted to document patterns of sea surface T (SST) [3,4,7] to such an extent that some studies concerning coral bleaching do not employ any in situ data, relying solely on SSST [8–12]. The 1998 La Niña warmed the waters around Palau and the western Pacific, which caused coral bleaching, high mortality, and the degradation of reefs [13]. In response to this event, a network of in situ T measurements was

initiated in Palau to understand SST variability and its effects on reefs (Figure 1). While SSST describes the broad spatial and temporal patterns, this network reveals large amplitude changes, on periods of minutes to years on the outer reef slopes, some of which are invisible to SSST. The overall Palau network presently comprises over 70 stations and 150 compact T loggers (TL), which are deployed by divers [14]. This paper mainly offers guidance from experience on how to set up and maintain an extensive "value-added" T monitoring network in a coral reef area.

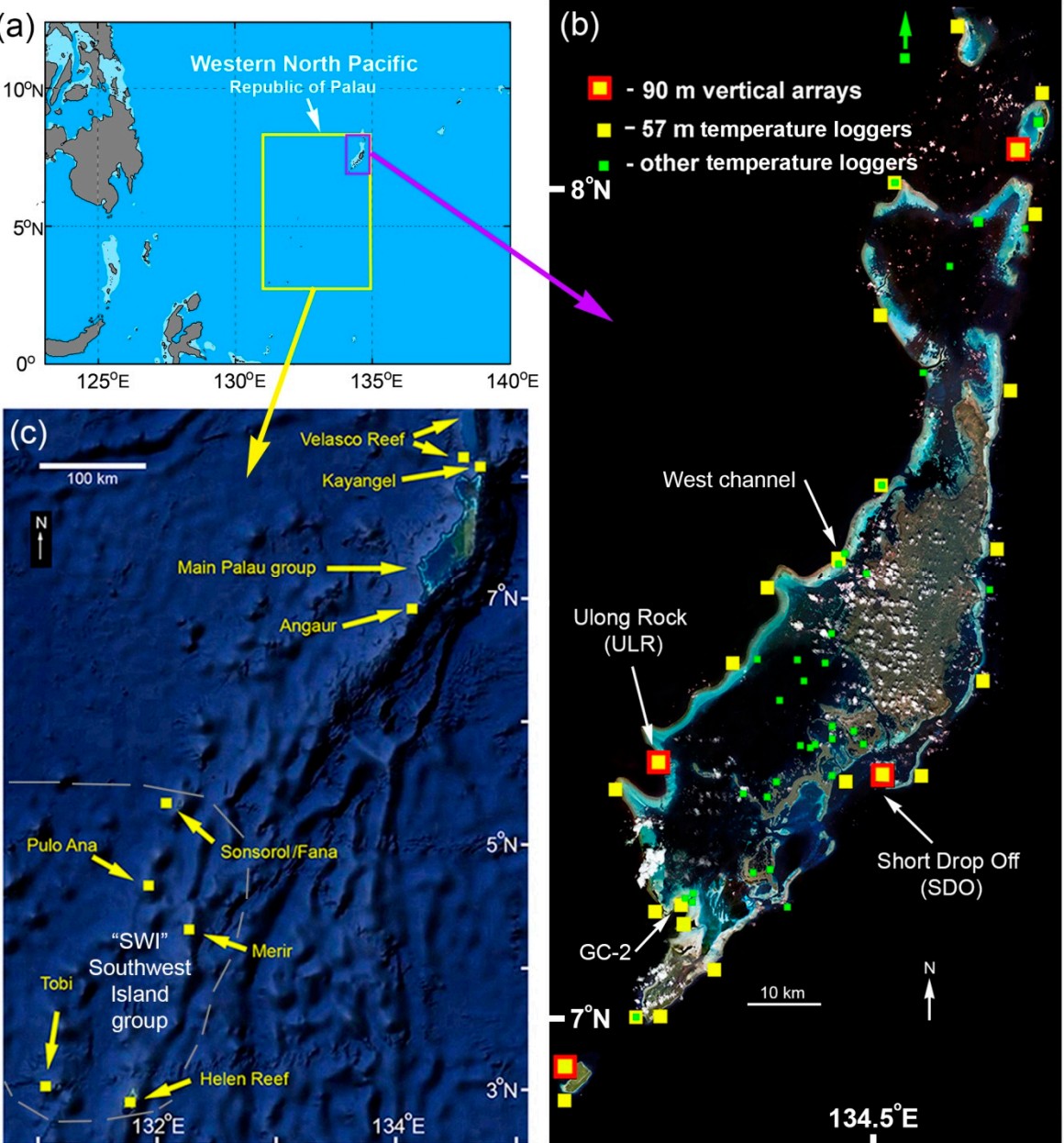

**Figure 1.** (**a**) General map of the western North Pacific with the location of Palau indicated. The areas shown in (**b**) and (**c**) are indicated by purple and yellow boxes respectively. (**b**) Locations of main temperature monitoring arrays and stations (Ulong Rock, Short Drop Off, GC-2 German Channel, Western Channel) on the outer slope and lagoon regions of Palau. (**c**) Locations of temperature logger stations in outlying areas of Palau, including the Southwest Islands group (SWI) and northern Ngeruangel/Velasco Reef area.

Three concerns about SSST versus in situ data belies uncritical acceptance of the implicit assumption: (1) accuracy compared to shallow (less than 5 to 10 m depth) in situ conditions; (2) diurnal T changes

occurring in the upper few meters of the water column; (3) vertical and horizontal variation in thermal conditions. SSST is determined via emitted infrared radiation from the uppermost 20 μm of the water column ("SSTskin") with a 1 to 4 km horizontal resolution and microwave radiation from the upper 1 mm sub-skin ("SSTsubskin") with a circa 25 km resolution [15]. T immediately below these thin layers ("near-surface SST") is not measured by satellite but comes principally from drifting buoys (Global Drifter Program, 20 cm probe depth) and shipboard measurements (various shallow depths of 1 m or more) and are then used to validate SSST data [2]. Oceanic moorings with T probes along their length, such as TAO/TRITON (the nearest to Palau is 260 km away) and ARGO profiling floats (which do not regularly measure T above 5 to 10 m depth) do provide some data at coral reef depths, but almost always in the open ocean far distant from reefs [16]).

The diurnal daytime T changes in the upper few meters of the water column are "visible" to satellites, but not captured as SSST. Solar heating warms water over very shallow reefs [14] with minimum daily SSST measured near dawn [17]. Diurnal patterns expose reefs to higher potential heat stress during the day and important heat content information is lost using only nighttime data. Very shallow stratification can occur, particularly with low wind speeds and calm seas, while wind, waves, and tidal currents can increase shallow mixing and advection of heat at other times. At some depth below the surface, usually a few to several meters, the diurnal heating signal is largely lost, and T remains diurnally stable [14], often to depths of tens of meters. In summary, [15] provide a similar framework for discussing SST defining SSTskin (upper 20 μm), SSTsubskin (upper 1 mm), SSTdepth (about 10 m), and SSTfoundation > 10 m). For our purposes, we refer to SSTskin and SST subskin as SSST. SSTdepth has diurnal variation and SSTfoundation has none. To make sense of these data, environmental conditions at the time of acquisition must be taken into account. On and near reefs, these conditions show extreme variability in periods ranging from minutes to decades, as we show below, some of which cannot be detected in SSST. We focus mainly on SSTdepth and SSTfoundation.

Photophilic reefs occur from the shallows (upward limits determined by sea level) to over 100 m depth in clear tropical waters [18]. Detailed measurement/analysis of thermal patterns over depth ranges and time across a reef tract indicate SSST has a limited capacity to capture the dynamics in the environment [19–21]. On outer slopes once below the diurnally variable depths (usually a few to several meters), "foundation" T in most tropical-subtropical reefs is relatively stable for a few tens of meters. Descending further, however, T (and light levels) decreases in the deepest areas of reefs experiencing consistently lower T, all invisible to satellites. Moving horizontally from ocean slope to inner lagoon or channel areas T patterns vary, but inshore areas are generally warmer. Tidal currents advect water from the ocean to a lagoon (or the reverse) moving shallow water masses of different T/salinity across environments. Adjacent landmasses may influence T in lagoons or over fringing reefs. Even the 5 km pixel size of thermal remote sensing [22] still limits the ability to measure skin T in many areas due to their geography.

The vertical distribution of T over different time scales (decades to minutes) can differ considerably from SSST. During La Niña, Palau's reef waters are warmest. In addition, SSST and 2 m depth in situ values are relatively close, but correlations decrease for in situ data at 15 and 35 m [14]. During ENSO neutral conditions, SSST correlations with in situ T decrease at all depths compared to La Niña. During El Niño, the thermocline shoals and cool water is brought closer to the surface. Surface waters remain relatively warm but the correlation of SSTdepth and deeper in situ data are low, while SSST can be several degrees warmer. Near the lower depth limits of photophilic reefs in Palau (roughly 60 to 70 m) differences between SSST and in situ data are even larger (as much as 12 to 15 °C), with high daily variation in T over both short (min to hours) and long periods (weeks to years, ENSO related).

Accepting that SSST is not a panacea for all consideration of T on coral reefs and that the "implicit assumption" may not be universally applicable, increased in situ measurements are needed to understand the scale of differences in thermal conditions on broad reef areas over depth and time. Only accurate in situ reef T data, the "gold standard" [14], can inform researchers and managers regarding how representative are SSST data and to move beyond generalized SSST values to understand

that each locality and habitat is different. For most reef areas, a broad reach of monitoring efforts (temporal, vertical, horizontal) is needed to document representative conditions.

## 2. Materials and Methods

### 2.1. Palau and It's Monitoring Network

The Republic of Palau is a western Pacific country with extensive and diverse reef systems located at 3° to 8° N (Figure 1). An extensive barrier/fringing reef (300 km in length) surrounds the main island group. Several lagoons total nearly 1000 km$^2$ in area with many reefs [23] inside the barrier reef. Reefs are found along or close to the shores of islands, producing close interactions between reefs and land masses. Within 50 km of the main Palau reef tract, an oceanic island (Angaur), a small atoll (Kayangel) and a large sunken atoll (Ngeruangel/Velasco Reef-area 400 km$^2$) occur. Five oceanic islands and one atoll, collectively called the Southwest Islands (SWI) occur 300 to 500 km southwest from the main reef/island complex (Figure 1c). Found at 3° to 5° N, the SWI are usually within the flow of the eastward North Equatorial Counter Current (NECC). The main group, centered on 7° N, lies at the southern edge of the westward North Equatorial Current (NEC) but is impacted by the NECC as it shifts northward seasonally and during ENSO.

Palau is ideal for examining tropical upper ocean T variability over broad to fine spatial and temporal scales. It has a narrow annual shallow-water T range (1.5 to 2.5 °C), but high variation (over 15 °C) in daily/weekly means in the deeper photic zone. It experienced a major coral bleaching event in 1998 [13], a lesser event in 2010 [12], and a series of small localized events in other years [23]. In 1998, excessively warm water caused loss of zooxanthellae (bleaching) in reef corals resulting in high mortality and degradation throughout the depth range of reefs [13]. At that time there was no accurate T data for Palau's reefs, but in the wake of that event efforts to document Palau's ocean T were started. In 1999, an outer reef vertical array of 5 TLs at depths of 2, 15, 35, 57, and 90 m was set up at Short Drop Off (SDO) on the eastern barrier reef (Figure 1b). Concurrently, a few TLs were installed on shallow inshore/lagoon reefs. Initial array results indicated rapid and extreme variations at the 57 and 90 m stations on the profile. In 2000, a second equivalent array was established at "Ulong Rock" (ULR) on the western barrier reef [24], which recorded some of the largest isothermal vertical displacements ever. Since then the network has expanded to over 70 stations with 150 TLs, ranging from all of Palau's SWI to the north reefs (Ngerunagel/Velasco Reef) with broad coverage of habitats and depths within lagoon areas.

To look at deep reef internal waves, a new initiative deployed TLs in 2014 at 57 m depth at 27 stations, which were widely distributed on the outer slopes of the main Palau reef tract, Kayangel Atoll, Angaur Island, and Velasco Reef. Called the "deep network", they record T once a min, each recording 525,600 measurements every regular year. The depth of 57 m was selected due to the high variability during both El Niño and La Niña periods seen in vertical array data and was a depth from which reasonably easy recoveries/deployments were possible by experienced divers. The deeper stations of the four 90 m depth vertical arrays along Palau's main reefs (Figure 1b) also switched to one-min interval data collection in 2013. In 2015, the deep network was extended to the SWI with five 57 m stations (now six), along with shallower loggers at each location. The deep network, as of January 2020, has recorded 110 million high-resolution T measurements while the remainder of the Palau network accounts for about 10 million more.

Do outer reef slope TL data accurately represent T profiles found in adjacent deep ocean? During 2009 to 2019, Spray autonomous glider missions for Office of Naval Research initiatives [25–27] gathered data on T. At the start and end of wide-ranging missions, data were collected within a few km of the reefs (surface to 200 to 500 m depth) for several days. There was close agreement between open ocean glider and outer slope TL data [28,29].

*2.2. Selection of Stations*

Care in selecting monitoring station sites is important, particularly where geomorphology or oceanographic conditions may influence thermal conditions. The initial outer reef (SDO and ULR) array sites simply added temperature monitoring to ongoing ecological work, but additional outer reefs stations were subsequently positioned to provide geographic/habitat diversity, such as adding stations on the North and South slope to supplement those on East and West. At some level station selection is "logical" but as a network develops, specific sites may be selected solely on the basis of having an interest in knowing of a detailed area, without knowing in advance whether that station will prove interesting (or not). Lagoon and island stations can be located in what are considered "typical" sites where results may be representative of similar habitats. If a truly distinctive habitat is present, stations can be located to delineate the specific conditions found there, which may differ from nearby areas.

Accessibility during most weather conditions and ease in locating them via distinctive surface/shallow features are advantageous. Each site, even with distinctive features, needs its position established by GPS to a resolution of 0.001' (circa 1.8 m) and recorded, allowing the site to be located if weather is poor and visibility limited. Sites can also be selected based on additional scientific interest, with other long-term studies being undertaken concurrently (e.g., photo transects) without the need for a separate field trip. T records from such study sites are useful when events (coral bleaching, storms) occur and thermal data are relevant. Stations located in broad shallow areas such as barrier reef tops, that may be accessible only at spring high tides, are often disturbed by storm events and may have few distinctive features (i.e., large coral heads) making relocation of TL instruments difficult. Hence, the need for greater care in choosing sites in these areas is implied and using all available methods to ensure that they can be located later for TL exchange.

*2.3. TLs and Their Mounting: Considerations Regarding Installation and Deployment of Instruments*

Four types of T loggers have been used: (1) Onset Hobo Pro-8; (2) Onset U22 loggers; (3) Seabird Electronics (SBE) 56; (4) RBR Solo loggers, all relatively small and lightweight. Four factors determined the preferred unit for a station: resolution/accuracy, sampling interval, memory capacity, and response time. In the present effort, target accuracy was 0.1 °C, with a higher precision of up to 0.01 °C with the SBE and RBR loggers. The target accuracy was probably a reasonable compromise compared to SSST data, which is also presented as 0.1 °C. Given the many aspects of variation at individual stations for bleaching indices, the existing accuracy is satisfactory especially considering some of the large amplitude signals.

For all TLs it is important to have a weight attached so it has extra negative buoyancy. The Onset U22 is positively buoyant, the RBR Solo near neutral, and the SBE 56 slightly negative. Instruments and attached weights were deployed, either fixed to the bottom or placed unattached (free) onto the bottom. Mounting instruments on a mooring with float and anchor is risky, as the unit may drift away if the mooring anchor fails, and not advised.

*2.4. Characteristics of Instruments*

1. Onset Hobo-Pro-8 (Onset Computer Corp., Bourne, MA, USA), units (used 1999 to 2006) recorded to 0.01 °C precision but had to be installed in water/pressure proof housings with only their cabled thermistor probe exposed. Other disadvantages were: limited memory (11 months at a 30 min sampling interval); limited battery capacity; the need to disassemble the housing to replace batteries and download data; continuous problems with leaking in the complicated housing/probe system. This caused some unacceptable loss of data in the early years of the program.

2. The Onset Water Temp Pro 2 (U22)(Onset Computer Corp., Bourne, MA, USA), is (2004 to present) pressure-proof to 120 m depth, has easy deployment, recovery, and download with batteries lasting several years over multiple deployments. With a 30-min sampling interval, the memory records for

two and a half years. Disadvantages are positive buoyancy (requires attachment to a weight), slow response time, a special adapter needed for deployment/download, and difficult battery replacement (return to manufacturer). Details are described subsequently. Cost: US $129.

3. The SBE 56 (Seabird Electronics, Bellevue, WA, USA) (2010 to present) has been reliable, but expensive (four to five times the cost of the Onset U22). Advantages are high resolution, quick response time (exposed probe), long battery life, large memory, and simple set-up and download. It has been used for deep-water stations where rapid thermal changes occur, as well as areas with one to three-year recovery/deployment intervals.

4. The RBR Solo (RBR Ltd., Ottawa, ON, Canada) is similar to the SBE 56 in capabilities/cost and is slightly shorter in length. Advantages are comparable to the SBE 56, but some data have been lost due to power (battery) glitches while deployed. This is presently used in "low priority" locations, where loss of data is not as dire, needing short sampling intervals.

5. Other oceanographic instruments (Onset pressure loggers (Onset Computer Corp., Bourne, MA, USA), Acoustic Doppler Current Profilers, wave gauges, conductivity T loggers) have provided additional T data to the program and their data are considered supplementary, not primary. In a few cases, fast logging dedicated T loggers were attached to longer interval sampling ADCPs or others. Direct data comparisons indicate they can be acceptable substitutes for dedicated TLs.

There are other brands and models of TLs available, but none were used in this project.

### 2.5. Calibration of TLs

New TLs were calibrated in a water bath at 20 to 31 °C before deployment to establish baseline accuracy against a NIST traceable mercury thermometer, as well as cross calibrating each with other new TLs and older units between deployments. Later re-calibration (between deployments) used similar methods. The target accuracy for all measurements was 0.1 °C.

For calibration, all TLs were set to start simultaneously with a 1-min interval. They were bundled together using rubber bands or similar, placed with probes down while immersed in a circulating water bath and equilibrated for 5 min or more prior to measurements. The mercury thermometer, its bulb situated at the level with the thermistors, was read using a magnifier to 0.02–0.03 °C resolution on the minute when TLs logged values. Calibration runs lasted 30 to 120 min at various T in the water bath. After downloading, data were entered into a spreadsheet comparing each TL against all others in the run and the mercury thermometer. A correction factor was calculated for each logger and applied if needed.

### 2.6. Sampling Intervals

The measurement interval is a compromise between memory size, battery life, and desired temporal resolution. For most stations with depths less than 35 m, a 30 min interval was used. An Onset U22 with a new battery would have adequate battery life for five to six years at that interval, thus although memory-limited, a new unit could be deployed for two successive periods of 2+ years. In a few instances where Onset U-22 TLs needed to be deployed for over two and a half years, a 1-hr interval was selected to extend memory duration [30].

For thermally active stations SBE 56 or RBR TLs sampled at 1-min intervals. An Onset U-22 sampling at this interval would fill its memory in only 30 days, thus such short sampling intervals were only practical with the more expensive SBE and RBR loggers. At stations where only minor variation in T is anticipated, deployment of high-resolution SBE and RBR TLs may be appropriate if there is a need to verify low variation.

Software for TLs (Hoboware 3.7.18, Onset Computer Corp., Bourne, MA, USA; Seaterm V2, Seabird Electronics, Bellevue, WA, USA; Ruskin 2.10.4, RBR Ltd., Ottawa, ON, Canada) allowed for delayed starts and if several TLs are prepared for array deployment, they should be set to start logging at the same time. Data logged prior to deployment or after recovery should eventually be deleted after download and carefully noting times of first/last good samples (the time the unit is at the proper depth,

not just in the water) will allow data to be deleted. Often shallow TLs were recovered while descending to retrieve deeper units, and logged low Ts on such excursions, which needed to be deleted afterward.

## 2.7. Preparation of Instruments for Deployment

**Onset U22.** If deployed/recovered by a diver they are best mounted on individual 1 to 2 kg weights, such as the lead weights used for SCUBA diving (Figure 2a). A vinyl plastic cap is placed over the wide "download" end of the TL, to protect the clear plastic window, and the remainder of the TL wrapped in plastic or electrical tape to reduce biofouling. The thermistor is inside the housing near the mounting lug of the housing and this area should be exposed to the water to retain response time.

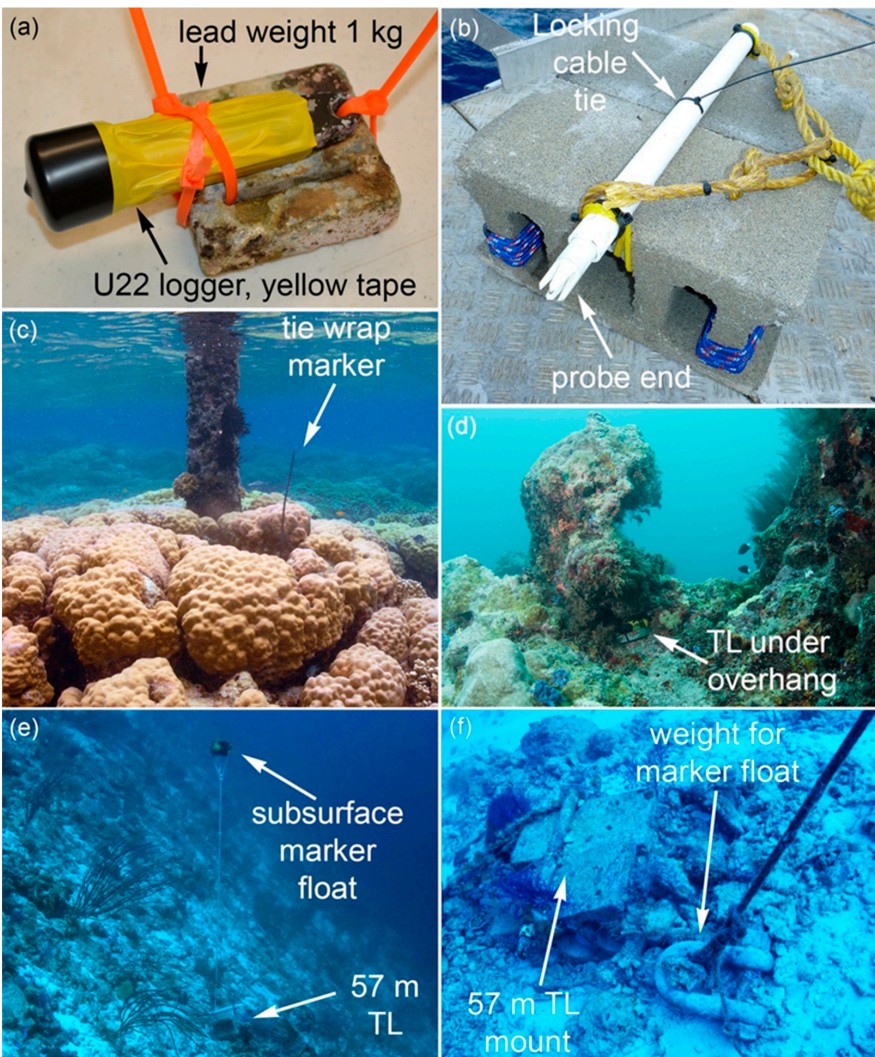

**Figure 2.** Various TL deployment methods. (**a**) Onset U22 mounted on dive weight with cable ties. (**b**) Deep network (57 m) mount with a tube for SBE or RBR TL, cable tie lock shown. The line shown used was only for lowering the mount to the bottom, then cut and removed. (**c**) Location of U22 temperature logger (TL) on a very shallow reef positioned in a crevice for protection from direct sunlight, and its location indicated by a large cable tie. (**d**) U22 TL on a reef, rarely (if ever) visited by divers, attached by cable tie and openly visible to aid in locating it for recovery. (**e**) Deep network station (T05) on a steep slope at 57 m depth with marker float to assist in locating the instrument. (**f**) T05 station after five years on the bottom, TL mounting tube attached to two concrete blocks. The logger was exchanged each year for a fresh unit. A marker float is on a line with separate weight (a 15 kg shackle) and not attached to the TL mounting.

One way to install the TL on a weight is to drill a 1/4 to 3/8-inch diameter hole in one margin of the weight. The TL, wrapped in tape with plastic cap in place, is then attached using three plastic cable ties (also known as "zip ties"); one going through both the hole drilled in the weight and the mounting lug of the TL, while two others overlapping diagonally are used to secure the TL at the opposite corners of the weight (Figure 2a). If drilling a hole is inconvenient, additional cable ties help to secure the TL to the weight. Cable ties are readily available, inexpensive, and easily cut if needed. It is tempting to simply attach Onset U-22 loggers to the bottom using a cable tie or similar through the mounting lug and then attaching that to something solid on the bottom. The Onset U22, however, is positively buoyant and if the cable tie fails (which occurs surprisingly often with upward force on the latching mechanism), the unit will drift away. If secured to a weight by at least three cable ties, it is improbable for all to fail and the unit to drift away.

**SBE 56 and RBR Solo.** These cylindrical TLs have a 25 mm outside diameter, fitting inside a 12 inch (30 cm) long piece of 1-inch (25 mm) inside diameter PVC plastic pipe. The bore of the pipe is usually slightly oversized inside (26 mm), a close fit for the TL but still with some clearance. The TLs, a mounting lug with a 5/16" (8 mm) diameter hole on one end and exposed probe on the other, can be inserted inside the pipe to a level so only the probe end of the TL is exposed (Figure 2b). Holes of 3/8 to 1/2" diameter (10 to 13 mm) pre-drilled through the PVC pipe at the correct distance allowed a single cable tie to be inserted through the tube walls and mounting lug, then secured back on itself, to lock the TL into the tube. If holes in the tube for the cable tie "lock" are too small (1/4" or 6 mm diameter), this can make inserting the cable tie at depth difficult. The TL can be exchanged at depth by a diver cutting and removing the single cable tie, pulling the TL out of the tube, inserting a new unit to the correct level, then inserting new cable tie and locking in place. The SBE and RBR TLs are of different lengths and require holes at different distances along the tube (19/26 cm) to have just the probe exposed. When preparing mounting tubes, it is convenient to drill holes at the correct distance for either type of TLs, so tubes are interchangeable.

A completed mounting tube (with or without the TL already installed) can be attached to a 1 to 2 kg weight or another object for securing on the bottom. For the "deep network", standard mountings were made from two concrete blocks tied securely together and with the mounting tube permanently installed on one surface (Figure 2b,f). The blocks and tube are not recovered, instead the diver exchanges the TL by cutting the retaining cable tie, pulling out the TL, inserting a new TL to the proper level in the tube, and then inserting and locking in place with a new cable tie. With practice, this can be accomplished on the bottom in less than 1 min. The recovered TL is placed in a mesh bag carried by the diver and brought up. The time of recovery to the minute should be noted to establish "last good" and "first good" values for the deployment. Mounting instruments on moorings consisting of an anchor weight (or line tied to the reef), line and float is not recommended. They can easily drift away if the mooring fails. In some cases, marker floats are helpful in finding TLs, particularly if diving from a shot line (Figure 2e), but if marker floats are desired, they should be installed separate (unattached) from the TL itself (Figure 2f).

## 2.8. Divers and Diving Deployments

Nearly all TL deployments were done by free-swimming divers. To depths of 35 m, TLs were readily deployed by recreational-level trained divers. Using nitrox (oxygen-enriched air), instead of compressed air, allows extended bottom times at or quicker repetitive dives to 30 to 35 m depth. Deeper deployments and exchanges were accomplished using deep-air diving (to 60 m) or trimix (helium/air) gases (60 to 95 m depth), requiring specialized training and equipment. Such "technical diving" has become increasingly common, and given the proper equipment, training, and planning, it is feasible to diver deploy/recover TLs to depths of 100 m [31]. In Palau, there were numerous individuals, usually working in the tourism dive industry, who were capable of doing this type of diving; not an unusual situation considering the worldwide development of diving tourism.

### 2.8.1. Shallow Deployments

"Shallow" TL deployments are considered to range from the surface layer to the depths of 30 to 40 m with no decompression diving. Across this range, different conditions are encountered, and deployments optimized for the conditions expected. A primary consideration when deploying is the ease with which a TL can be located and recovered months to years later. Over time it may become covered in marine growth or the structure to which it was attached crumbled or gone.

TLs secured to a small weight may be put out on the reef, either free on the bottom (not secured, but in a location where their weight makes it unlikely to be moved by currents or waves) or attached to the bottom. The use of SCUBA weights for TLs has the advantage that the belt loops designed into them are convenient openings through which to run cable ties to secure them to the bottom (Figure 2a). Cable ties come in various lengths, making it easy to go around large objects or to thread through small openings for securing TLs (Figure 2d). Often an area of rocky reef will have small rock arches where the TL can be secured by a long cable tie.

In some cases, it is easier not to attach a weight/logger to the bottom. If the unit is located in a slight depression where being moved is unlikely, it can just be dropped off without further attachment. Nevertheless, ideally all TLs should be attached to the bottom, even if only loosely, since if left unattached and it disappears, it is necessary to search around the site (often difficult at depth) and if not found, no reason can truly be ascribed to its loss. The trade-off is that it takes longer to deploy and recover the unit and in deep water seconds are important in this process.

### 2.8.2. Special Considerations for Very Shallow Sites

Sites less than 5 m deep should be considered "very shallow" and in areas exposed to waves and swell, the TL must be mounted in a manner capable of surviving very rough conditions. TLs securely attached to weights, still need to be attached as a unit to a strong point (rock) on the reef itself. Use of heavy-duty cable ties (strap width 3/8 inches or more, with a breaking strength of 100 kg or more) is the easiest method for securing units in rough areas. Multiple cable ties should be used, and efforts made to ensure the unit once secured cannot be moved, otherwise, over time the unit may work loose and be potentially lost.

### 2.8.3. Shelter for Loggers

In very shallow water, direct sunlight may heat a logger resulting in values in excess of in situ water Ts with an observed an increase of 1 to 3 °C for loggers (depending on model) submerged in a flow-through mesocosm, but openly exposed to the sun [32]. It is recommended to either place the logger in a location sheltered from direct sunlight or install a reflective shade over the unit.

### 2.8.4. Marking Sites

Whether or not to hide TLs on the reef is an important consideration. For shallow water deployments in areas frequented by divers or fishers using SCUBA/snorkeling equipment, it is critical to hide loggers so that they are not seen and picked up inadvertently. Once coated in biofouling, a TL often looks like something that has been lost, perhaps dropped from a boat; not something that has been put in place for scientific research. When TLs are hidden, tucked into crevices or beneath overhangs, it is important to be able to find them again. Marking the general site where a TL is located (but not the TL itself) is helpful, with cable ties attached to a rock or projection nearby as an easy solution (Figure 2c). Photographs of the area (Figure 2d) are also very helpful, covering the area nearby, showing marking cable ties or showing locations of features, which are distinctive to help identify locations of TLs.

In areas where someone finding and picking up a TL is unlikely, efforts can be directed instead at making the TLs more visible underwater and easier to find. Attaching a number of long cable ties (30 to 60 cm), either to the weight or in the areas of the TL, so they stick out in various directions are

occasionally helpful in relocating TLs that have been deployed for some time (Figure 2c). The straight tails of the cable ties sticking out are distinctive, even if covered in algae. At times, cable tie tails sticking out of sediment from a buried TL has been the only portion of a TL visible and allowed recovery, otherwise the TL would have been lost.

Covering the TL in colored tape reduces biofouling and helps locate it (yellow is particularly distinctive; Figure 2a). Marking shallow sites with small floats aids visibility at some distance underwater. Careful notes should be taken on the nature of sites, times of deployment, exact depths (using a digital dive computer), and any other features that would help in recovery later.

### 2.8.5. Photos of Sites

Underwater photos of the sites, particularly showing the exact location where TLs are deployed, are important. Ideally, the TL should be visible in some photos (Figure 2d), to provide clues as to its location after it might be hidden in the reef. Underwater photo trails have proven particularly useful for locating sites on steep reef faces, with a shallow water (5 to 15 m depth) marker (sub-surface float, long cable ties) as a start point. A GoPro or similar camera taking time-lapse photos at 2-sec intervals helps document the routes swum when deploying the unit and can often be retraced if needed all the way to the location of the TL. This technique is especially helpful if the diver recovering the TL did not do the deployment.

### 2.8.6. Diving for Shallow Reef Deployments

If deployments are needed at depths of 30 to 40 m (the lower limits for "sport diving"), nitrox diving is advantageous as it allows increased bottom time (15 to 20 min) without decompression. If TLs are to be deployed/recovered at various depths moving along transects up a reef slope, the deepest should be done first, then move into shallower water. If doing a number of "swap-outs" of TLs, recovering old ones and setting out new ones, divers should have two mesh "dive bags"; one for recovered TLs and the second for ones to be deployed. The ones being deployed should be labeled indicating depth for each one to avoid confusion while diving. This prevents mistaking a just recovered TL for one that needs to be deployed, which is easy to do in the rush of a working dive.

### 2.8.7. Deeper Deployments

A number of countries, such as Australia and the United States, have regulations for the workplace and scientific diving which prevent (or make onerous) accessing depths below sport diving limits. Some offer dispensation to undertake advanced diving for research purposes [18]. In Palau, there were no regulations restricting experienced/trained individuals from undertaking the necessary dives to service the TL networks.

Careful selection of areas for deep deployments can simplify the diving involved, particularly for vertical arrays of several instruments. Near-vertical reef profiles are preferred, as this avoids divers having to make time-consuming horizontal or sloping swims between different depth stations. Reef markers, such as very long cable ties or subsurface floats, can be used to mark the route down the slope and individual stations to simplify finding TLs at depth. At the four vertical array sites to 90 m depth (Figure 1b), an experienced diver can exchange all TLs with only a short decompression. Obviously, extra decompression should be done for safety, but the decompression obligation with an efficient dive is not great. It is still incumbent on all persons using these techniques to understand the risks and difficulties involved, prior to attempting what are still very serious dives with risks from decompression sickness (bends), gas toxicity, and drowning. In many cases, scientists may prefer to have diving professionals do deep TL deployments. There is no operational need that would prevent such well-trained competent divers from doing so. As technical diving has become more common, in many locations advanced divers using mixed gas rebreathers can deploy and recover deep TLs.

### 2.8.8. Shot Line Diving

In some locations, the bottom will be sloping to such an extent that descending and ascending along that slope is not practical as it would require excessive bottom time at depth. In such a situation, the preferred technique is to dive using a "shot-line", an anchor weight (5 to 15 kg) with a line sufficiently long enough to reach the surface from the water depth and a large surface buoy as a position reference. The anchor weight is connected to the line by a clip, which can easily detach the line from the weight, so divers can potentially use the line for ascent leaving the weight on the bottom. When diving on a "shot line" the boat is not anchored ("live boat") but would hold station near the surface float during the dive. Assuming an accurate GPS position is known for the site, the boat slowly approaches the position and once directly over it, the shot line anchor is dropped, and the line is allowed to run free as the weight descends vertically to the bottom. Once on the bottom the remainder of the line and the float(s) are thrown overboard, so the shot line is free of the boat. Ideally, this should be done in conjunction with a depth sounder in the boat, to verify that the sounder depth agrees with the known depth of the TL. Once established on the bottom, the shot line float will indicate if there are currents by leaving a wake behind it, and if excessive, they may drag the anchor over the bottom. The location of the buoy should be monitored for several mins to make sure it is not being dragged over the bottom by currents (which would mean it is no longer at the location of the TL), then the divers prepare and dive.

The divers enter the water at the surface float and swim downward in constant sight of the line. Once near the bottom, they look for the TL location, which should be no more than a few meters away. If a subsurface buoy marking the TL location has been installed earlier (Figure 2e), it may be visible some distance above the bottom and quickly guide the divers to the TL. If not immediately visible, divers search along the correct depth for the TL (assuming a slope).

After locating the TL and exchanging instruments, the divers ascend along the shot line. The weight is detached and either abandoned or sent to the surface with a lift bag. The divers then use the shot line for the ascent. The boat remains close by and once the shot line weight with lift bag comes to the surface, it is picked up and then the boat follows the divers via the surface float. Bubbles should be visible on the surface as well. Divers complete any decompression hanging on the line in mid-water and after surfacing, are picked up by the boat. This technique is efficient when properly used, but like all deep-diving methods, divers need to be carefully trained and completely comfortable with the techniques involved.

In situations where diving cannot be used for deep TL deployment and recovery, remotely operated vehicles [33] or submersibles [34] could be used. These represent a major escalation in costs with uncertain recovery prospects, making extremely difficult what is often a relatively easy process if done by properly trained divers using advanced diving techniques [31].

### 2.8.9. Deep Moorings

It is tempting to deploy a heavy anchor weight, with a TL near the bottom on a long float line (20 to 30 m for a 57 m deployment). Often such floats are lost and if attached to the anchor weight create extra drag via the mooring line that may pull the entire mooring some distance. Such losses occurred with a few of the original deep network deployments.

One exception to this generality was an extremely deep difficult deployment at Ngaraard Pinnacle [30]. Due to the depth (95 m) over a 100 m deep bottom, currents, and the need to ascend in open water, the TL had been installed on a short mooring line with a weight and depth resistant plastic float and dropped from a boat. To recover the instrument two years later the mooring line was cut by a diver below the TL, allowing the float to carry the TL to the surface (where it was recovered by a boat) abandoning the weight on the bottom. A replacement TL mooring was dropped at the same spot, again by boat, with eventual plans to recover it in the same manner.

2.8.10. Data Management and Availability

Loggers were downloaded using the relevant software and raw data files archived on multiple computers. The raw data, which covered variable periods of time, were saved as Ascii text (i.e., as comma-separated values) or Excel files for further use. The variable deployment length data were put into discrete annual spreadsheets for each station/depth at the appropriate time resolution. The spreadsheet calculated daily/weekly means, range, and standard deviation.

Data are not presently in an open-access database, as they are actively being worked upon prior to publication. Selected data are made available to researchers on request via the CRRF website, which has a catalog of data files (https://wtc.coralreefpalau.org/).

**3. Results**

Data from TL networks can be used to address general questions (T regime of a reef area) or focus on specific patterns or dynamics within a larger reef tract (e.g., coral bleaching and ENSO changes). Here we provide (1) analysis of thermal patterns and variance on the Palau reef tract; (2) examples of why such a network is useful; (3) how it can be focused on obtaining information on poorly known aspects of reef science. Some examples shown use data from the 2015 to 2016 El Niño, as it was an exceptionally dynamic period, but other years (such as 2010 with coral bleaching) would have been equally informative.

*3.1. Outer Reef Temperatures Over Time and Depth of "Coral Reef Conditions"*

A vertical array of TLs (i.e., at several depths along the outer reef slope; Figure 3a) shows that the vertical profile of T is an important determinant of maximum reef depth. All TLs recorded at the same interval with an appropriate temporal resolution to document change over time periods ranging from minutes to years (Figure 3b) [14]. The initial array deployments (Figure 1b) on the East and West reef slopes (Short Drop Off in 1999 and Ulong Rock in 2000, Figure 3c) quickly identified two variable temporal patterns on the lower slope. Long-term (month/years) patterns of T varied by about 13 °C, while short term variability (min/hours) was almost as large at about 8 °C (Figure 3d). Initially, a 30-min sampling interval was used due to memory and battery limitations. After 2014 with new TLs, intervals were shortened to 1-min providing evidence of significant variation beyond that measured with a 30-min interval. The deepest depth of the array at 90 m (due to reef slope morphology and diving limitations) proved definitive because T values at 90 m were often well below the accepted limits for coral reefs. Based on data from all depths, 60 to 70 m was established as the approximate lower depth limit of coral reefs is Palau [29].

If full 30-min or 1-min data (respectively 17,560 and 525,600 data points per regular year) for multiple depth TLs at a single station are plotted together, patterns of short and long-term changes over that year are often evident (Figure 3c). The 1-min interval, in particular, reveals new dynamic patterns, such as during the El Niño of 2015 to 2016 [35]. Early in 2016, Ts at 57 m depth were much cooler than at 15 m depth, with a difference of roughly 10 to 12 °C, indicating a highly stratified water column. In March 2016, Ts at 57 and 90 m depth started increasing, along with a lesser increase at 11 m depth, rapidly hitting peaks from June to July 2016 (Figure 3b). The shift over 10 weeks away from strong El Niño conditions produced a "quasi" coral bleaching event [14], which reversed in July with shifts in oceanic conditions [35].

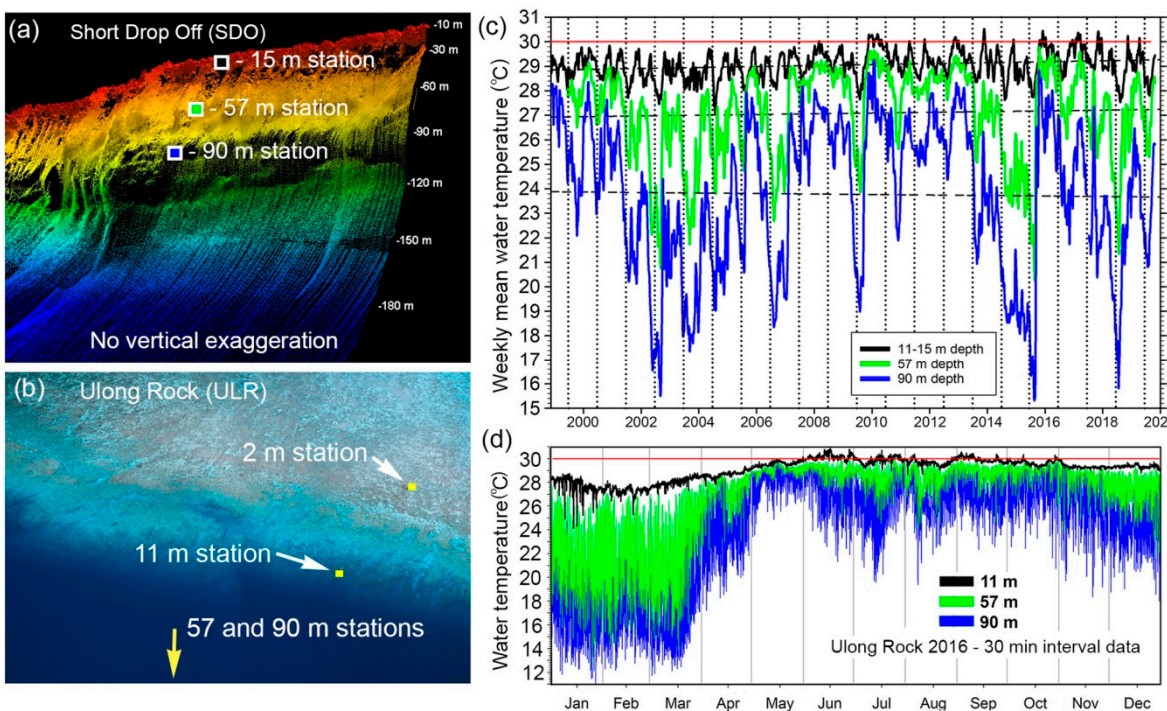

**Figure 3.** (**a**) Outer slope profile showing locations of TLs along the slope at Short Drop Off (SDO). The location of SDO is shown in Figure 1b. (**b**) Aerial view of Ulong Rock (ULR) station with general locations of TLs indicated, showing the horizontal displacement sometimes necessary to position TLs at certain depths. The location of ULR is shown in Figure 1b. (**c**) Weekly mean temperatures on outer reef slopes 1999 to 2020 in Palau. Only three nominal depths (15, 57, and 90 m) are shown; data also collected at additional depths. Nominal coral bleaching threshold of 30 °C is shown by the red line. Dashed lines are straight line regressions of all data at the three depths. (**d**) Year pattern for 2016 showing all 30-min interval data at the same depths as (**c**). Nominal 30 °C bleaching threshold is shown by the red line.

During La Niña, the deep reefs of Palau become extremely warm, with bleaching level T throughout the water column where reefs occur [14]. Mesophotic reef bleaching is largely unknown due to a lack of deep reef T data and surveys of bleaching at depth during times when bleaching conditions are present [18]. SSST provides no information to estimate deep bleaching. The linking of sea surface height (SSH) and SSST has the potential to estimate thermal conditions on deeper reefs [36,37], although this has not yet been incorporated into global bleaching estimations.

### 3.2. Impact of Internal Waves on Temperature Dynamics

Preliminary work documented the variable nature of deep T in Palau caused by internal waves/tides, suggesting 60 m depths were near the lower limits of photophilic reefs [24]. In 2014, the enhanced "deep network" was set up with 27 TLs recording at 1 min intervals at 57 m depth (Figure 1b), a depth with high thermal variation (Figures 1b and 3c,d). While this number of stations may seem excessive, preliminary data revealed that each station has a distinct short-term pattern of T variation but when all stations are considered together general patterns are apparent (e.g., a coherent diurnal internal wave in Figure 4). While a complete analysis has not been done, preliminary results indicate island-trapped internal waves can circulate part of the way around the outer slope [38].

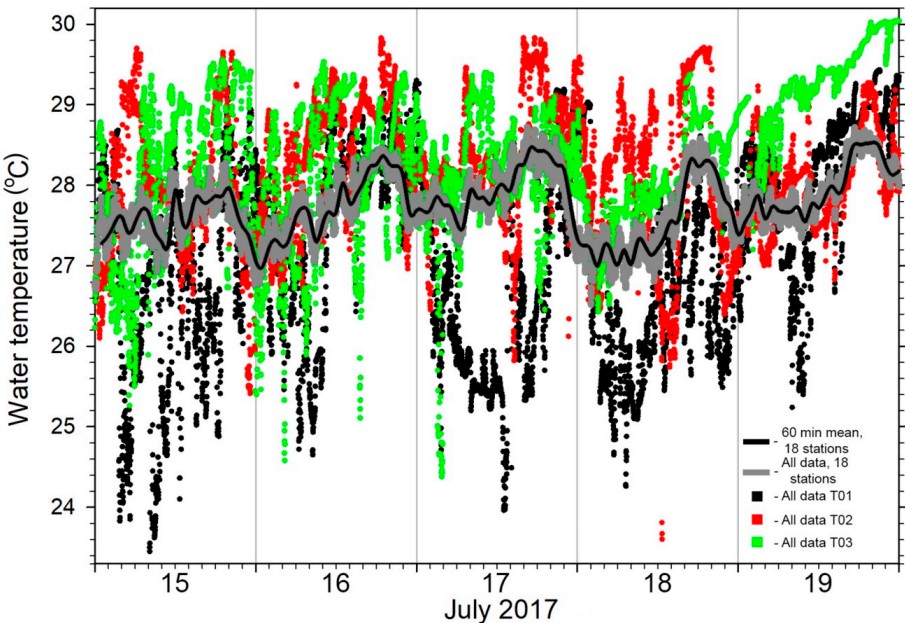

**Figure 4.** The "deep network" TLs at 57 m depth all around Palau measure once per min providing a wealth of data. This example shows over five days in July 2017, with the hourly mean values of 18 stations (those on the outer slope of the main group) (black line), the mean value each min for those stations (gray area), and examples of raw minute-by-minute data from three stations (T01-black, T02-red, T03-green) in the northern part of Palau indicating the high short-term variation at individual stations.

While most deep network stations are along steep (45° or more) slopes, a few are in areas with lower and more consistent slopes from deep water, where shoaling internal tides (internal waves forced by the semidiurnal or diurnal surface tides) are transformed into internal bores. A station at the South end of Angaur Island (Figure 5) had a remarkable T drop of 12.25 °C in one min (14.4 °C in 3 min). The generation and propagation of these waves are sensitive to stratification [39], which shifted dramatically during the end of the 2015 to 2016 El Niño [35].

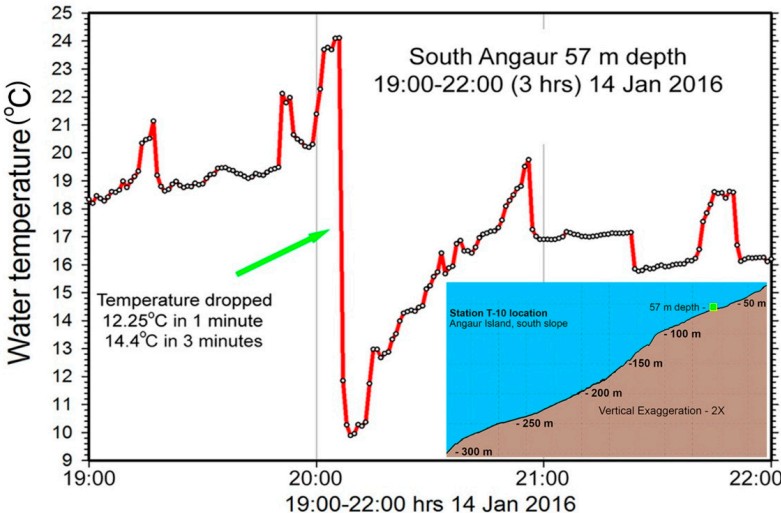

**Figure 5.** Extreme temperature changes at the southern end of Angaur Island, Station T10 at 57 m depth over three hours. The temperature dropped over 14 °C in three mins, although before and after this event, relatively normal variation in temperature at the station occurred. The gentle slope offshore of the South Angaur station is unusual for Palau's outer reefs (insert-lower right).

### 3.3. Mixing of Ocean and Lagoon Water while Advecting through Barrier Reef Channels

The deep channels bisecting the barrier reef between ocean and lagoon are likely sufficiently deep (35 to 75 m) to ingress water from ocean to lagoon that is vertically stratified, particularly during El Niño periods (Figure 6). Does this occur and how quickly would thermal stratification dissipate as water is mixed and moves into the lagoon? In 2010 a series of six stations with vertical TL arrays at 15, 30, 45, and 57 m (stations five to six without 57 m) depth were set up along the sides of the west/inner channel corridor west of Babeldaob Island (Figure 6a). At times thermally stratified water brought into the channel mouth penetrated several km into the lagoon on a diurnal cycle, but stratification vanished farther into the lagoon (Figure 6c) and water exiting the lagoon on falling tides was well mixed. In August 2010, a La Niña period, coral bleaching was occurring [12] and to a limited extent channels intermittently brought some cooler water into lagoons possibly due to a shoaling diurnal internal tide, which turns into a bore with steep/shallow isothermal slopes on the leading/trailing edge (Figure 6c). The potential effects of these processes on reefs are uncertain.

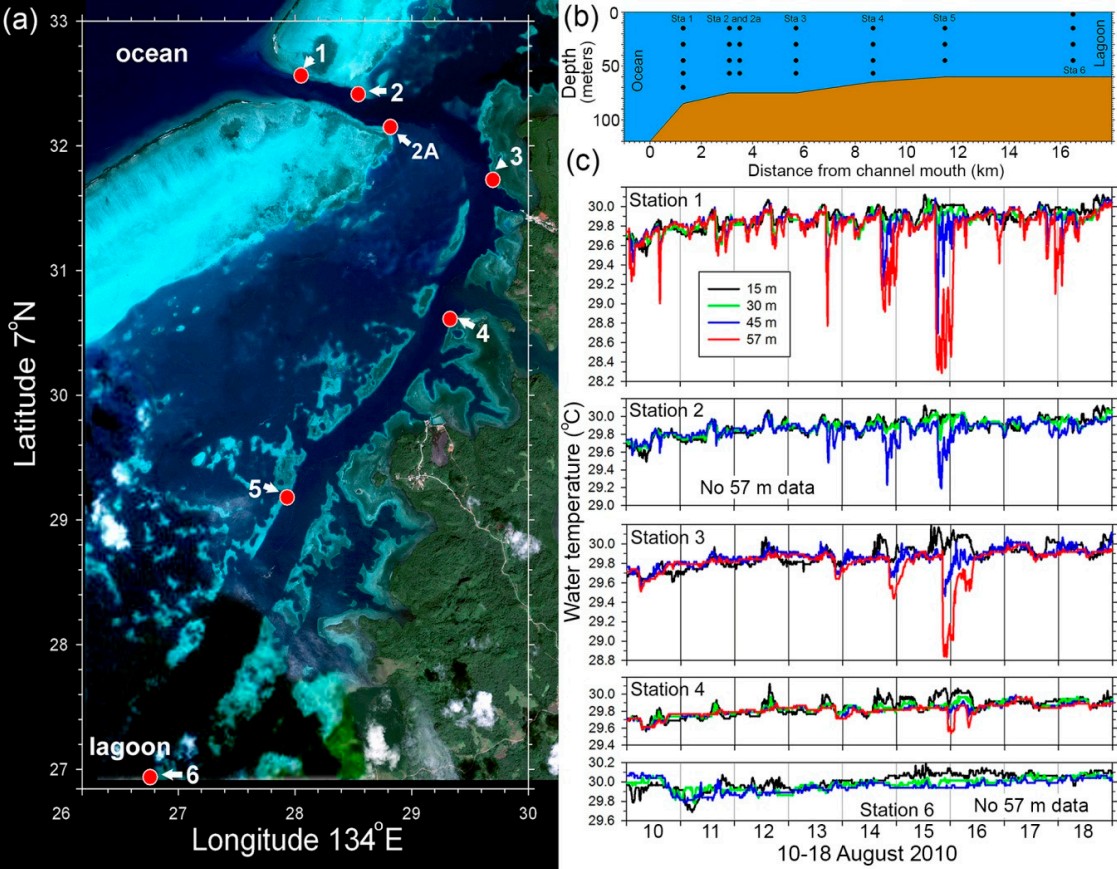

**Figure 6.** West Channel TL array. (**a**) Locations of stations with vertical arrays of TLs (indicated by numbers) along the sides of the West and Inner Channel into the lagoon. Minutes of latitude (7° N) and longitude (134° E) are indicated on the y- and x- axes. (**b**) Schematic of relative locations of numbered sampling stations along channel axis with the bottom depth of the channel shown. Black dots indicate depths of TLs. (**c**) Vertical structure of water temperature over nine days at stations along the West and Inner channels. Downward spikes in temperature are seen from 15 to 16 August 2010, with stations nearest the channel showing the largest decreases.

Different ENSO conditions change the nature of the offshore water column [28] and are reflected in the water brought into channels on rising tides. During the 2015 to 2016 El Niño water at 35 m depth on the slope of "German Channel" (GC-2 in Figure 1b) had quite variable Ts (Figure 7a). One year later, when the El Niño had dissipated, there was almost no variation in the Ts of incoming and outgoing

water (Figure 7b) and again the impacts of this variable T on channel reefs are unknown. Substantial variability is noted around Palau in observations from this network and elsewhere [40]. Furthermore, since these bores are turbulent [41] they can suspend and transport sediment, nutrients, or other properties into shallower water [42,43], which we have noted at this site or nearby.

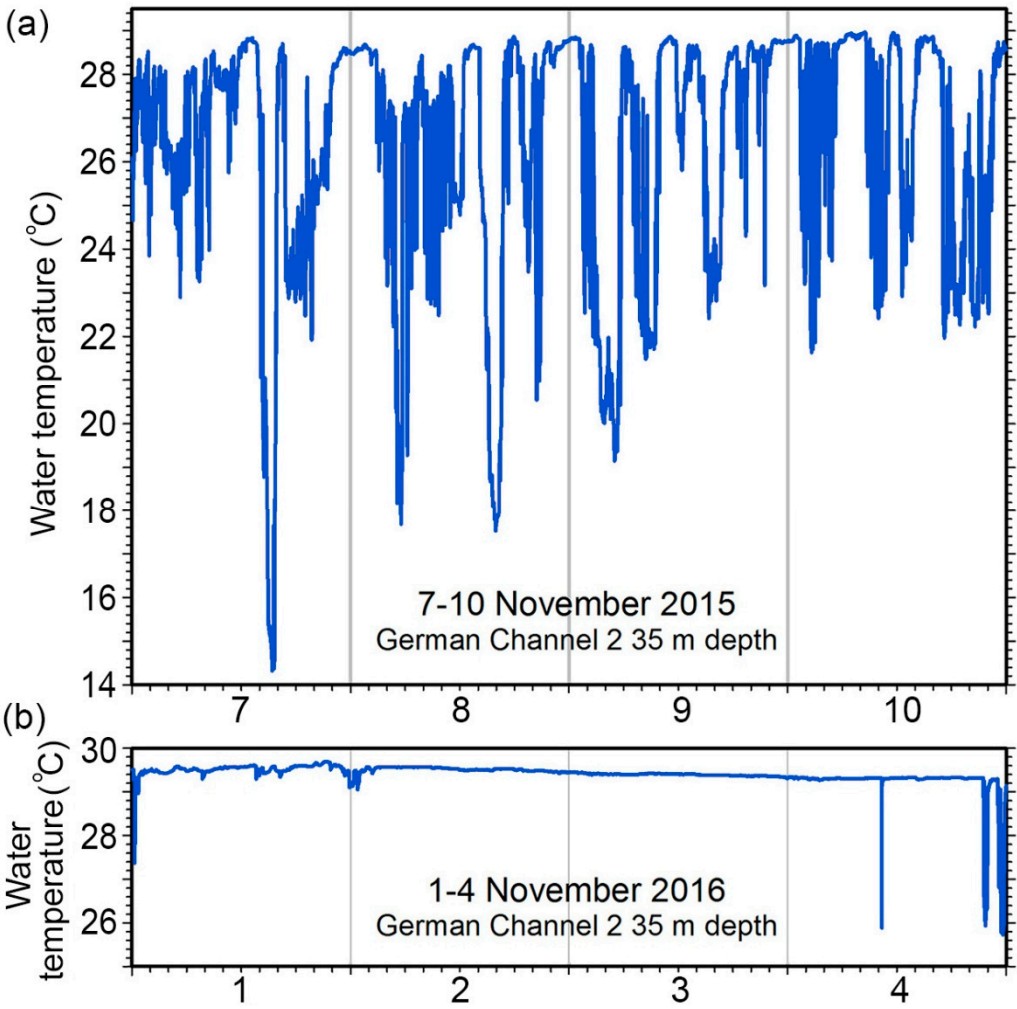

**Figure 7.** Temperatures at 35 m depth over four days on the slope of the German Channel station 2 (GC-2 location shown in Figure 1b) during (**a**) an El Niño period (7 to 10 November 2015) and (**b**) a period after the end of the El Niño (1 to 4 November 2016), one year later.

*3.4. Temperature Patterns across Broad Regions and Current Patterns*

With TL stations distributed over a wide geographic range (100 s of km), the network may capture differences attributable to broad oceanographic conditions. The SWI (Figure 1c) are within the eastward NECC, while the main island/reef group is normally dominated by the westward NEC, with occasional intrusions of the NECC. The main Palau group underwent a dramatic shift in currents, sea level, and T in 2016, while the SWI had a lesser shift in thermocline structure at the same time [35]. During the peak of the 2015 to 2016 El Niño in early 2016, shallow (11 to 15 m) daily mean T was very similar between Tobi and the main Palau group (Ulong Rock), separated by 600 km (Figure 8). However, at 57 m depth conditions were very different with Tobi near 25 °C while Ulong Rock was much cooler at 19 to 23 °C. As the El Niño ended in spring 2016, Ulong Rock had a major increase in T over 10 weeks. However, at Tobi T did not spike similarly, but started rising two months later (May) and more gradually; these large differences explained in terms of the dynamics of equatorial currents

and equatorial waves [35]. The result is that as the El Niño terminated, water moved back into the western Pacific, and forced the NECC northward towards the main Palau group.

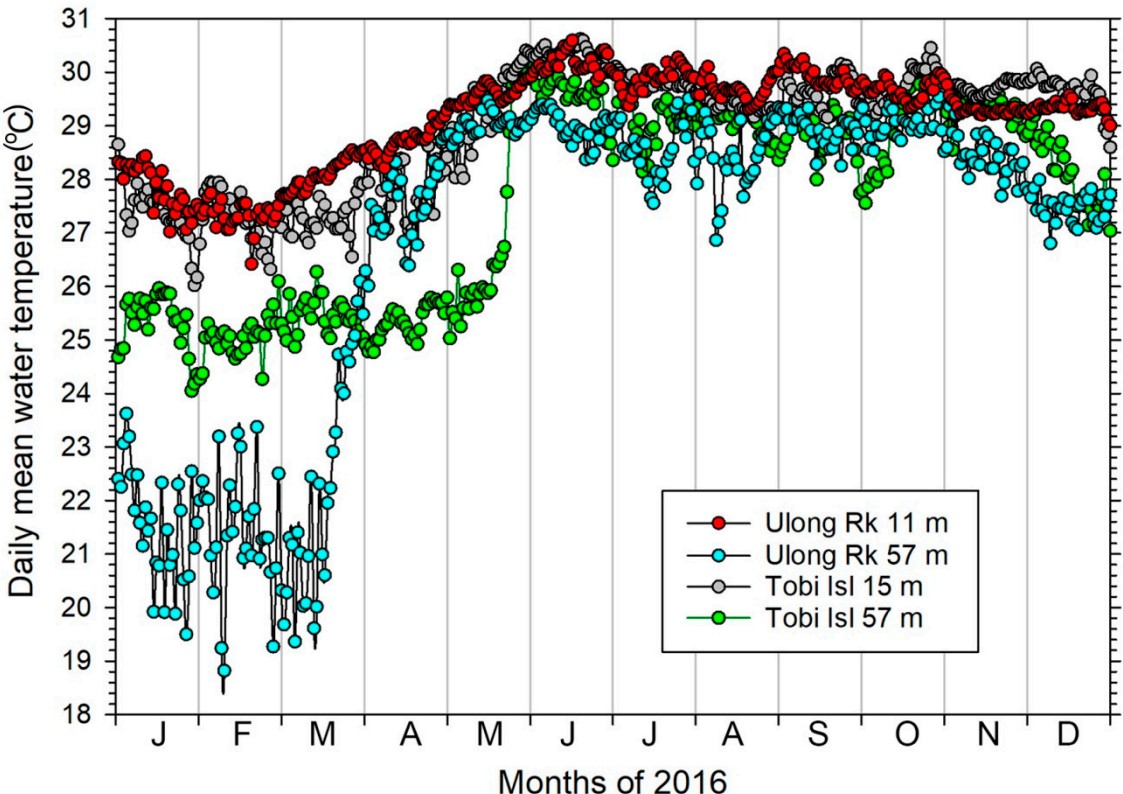

**Figure 8.** Daily mean temperatures at Tobi, Southwest Islands, and Ulong Rock, main Palau group at 15 and 57 m depths during 2016.

### 3.5. Climate Change Values from Long-Term Data

To identify climate change (decadal or longer-scale trends) long records are needed, which also resolve considerable variability from internal waves (minutes to hours). ENSO T shifts add another complication, with deeper areas having extreme variation. Periods of months to a few years clearly do not provide a sufficient length to observe whether a climate change signal is present.

With two decades of data from consistent locations, it is now possible to begin examining whether the data show trends potentially related to global climate change. The 11 m weekly mean data (30 min interval) from Ulong Rock ranges from below 27 to over 30 °C (Figure 9) while the 2 m data has a slightly greater range, reflective of diurnal variability in shallower depths. A nominal 30 °C "bleaching threshold" line (Figure 9, red line) [14] indicates this high T level has occurred during several years since 2007, but not between 1999 and 2007. Severe bleaching occurred in 1998, prior to the start of the T network, and temperatures were certainly above the "bleaching threshold". A straight-line regression from the data shows an upward trend of 0.4 °C over twenty years, or 0.2 °C per decade (Figure 9, green line). The rate of increase changes slightly as new data are added each successive year. The trend is about 0.1 °C per decade around Palau from 1971 to 2010 averaged from 0 to 700 m [44], while SST shows a trend of about 0.2 °C per decade from 1900 to 2008 [45]. If our measured trend of 0.2 °C per decade extends to 90 m, it still explains only a fraction (up to 30 mm per decade based on the thermal expansion of seawater) of the 1990s decade long change in sea level in Palau, which appears mainly due to trade wind intensification during that time [46].

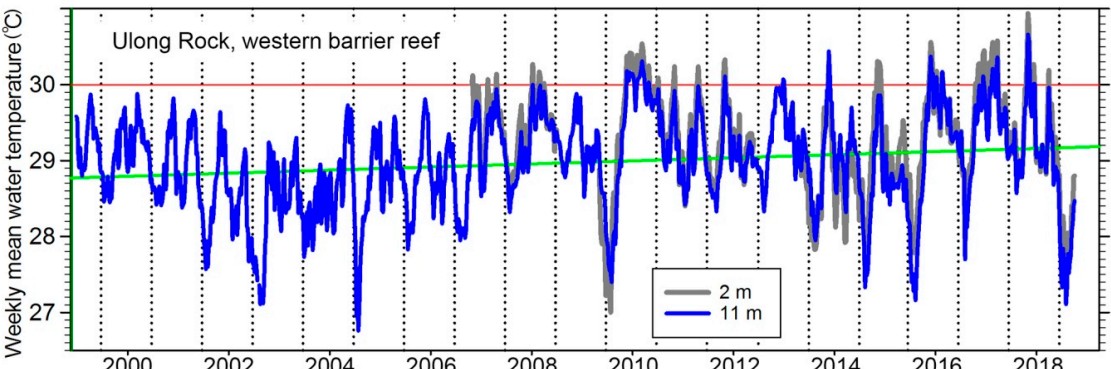

**Figure 9.** Weekly mean temperatures, 2 and 11 m, at Ulong Rock 2000 to 2019. The red line indicates a nominal 30 °C bleaching threshold above which a few weeks of exposure is associated with coral bleaching. The green is a straight-line regression of the 15 m data indicating a rise of about 0.2 °C per decade since 2000.

## 4. Discussion

### 4.1. SSST Versus in Situ Data

Both SSST and in situ data sets have their strengths. SSST provides the global perspective, but in situ data supplies most detail and serves as a ground truth for SSST. In terms of the calculation of heat stress and other T related parameters on reefs, accurate in situ are superior (and essential) when looking at a specific reef or site because of considerable horizontal, vertical, and temporal variability, some of which is invisible to SSST. Monthly mean climatology (MMC) based on SSST is used to define threshold Ts for coral bleaching and calculation of Degree Heating Weeks (DHW), an index of heat stress in a reef area to predict bleaching. If MMC and DHW are calculated from in situ data, the values are different, and one of the strengths of in situ data is that these indices can be determined for locations and depths, not just for single generalized location at the skin surface.

Detailed examination of T and coral bleaching has not been included here, although work has revealed without question the presence of thermal conditions inducing bleaching to 90 m depth in Palau. Deep (mesophotic) reef T conditions and subsequent coral bleaching has been largely ignored in the Indo-West Pacific [18]. Some areas, such as Australia, have experienced severe shallow bleaching events, particularly related to the 2015 to 16 El Niño, and are beginning to examine deep reef Ts/bleaching below 20 to 25 m [33]. While SSST alone cannot indicate Ts at the lower depths where reefs occur, the merging of sea surface height (SSH) data, either from satellites or tide gauges, with SSST provides a new way to assess heat stress in reef environments [36,38] and opens a remote sensing window into events at the lower limits of reefs. While this method required in situ data for validation, it can be expanded to other locations in the tropical Pacific where we expect the relation between Ts and SSH holds.

### 4.2. Why Is a Network with Many Instruments Needed in Coral Reef Areas?

The Palau network in the main island group provides about 54,000 discrete location–depth–time measurements per day while the NOAA Reef Watch virtual station data for Palau, https://coralreefwatch. noaa.gov/product/vs/data/palau, provides one daily set of SSST measurements (mean, maximum, and minimum), thus cannot capture the thermal dynamics within a reef area, particularly where ENSO related changes are large, internal wave variability is considerable, and diverse types of habitats are present. Without a network appropriately sized for the area to be covered, unknown aspects of the physical environment, along with the biological implications, will be invisible. Each of the vertical TL arrays (four at 2 to 90 m depth, several others at 2 to 57 m), plus the dozens of the widely distributed single 57 m TLs, have shown different patterns. It is not yet clear what is driving these differences, but the ocean current dynamics, documented recently [27,47] and others in the

same volume, as well as smaller-scale effects [29], produce a complex physical environment which continues to reveal new layers of complexity. The passages between island groups, such as between Peleliu/Angaur (Lukes Passage) and Kayangel/Ngeruangel (Velasco Reef) and the northern reef tract of the main group (Euchelel Ngeruangel, Kekerel Euchel) are exceptionally dynamic and influence the reefs in those areas greatly. Wake eddies and internal lee waves at 1-km scales are noted at points and over submarine ridges [48–50]. The seasonal shifts of the NEC and NECC, as well as with ENSO, are exceptionally important, changing the nature of the oceanic environment throughout Palau [35]. The same applies to ENSO cycles in the open ocean directly impacting reefs through impressive shifts in the conditions in the photic zone, poorly documented for western tropical Pacific waters [14,25].

The examples presented are largely concerned with outer reef vertical and temporal changes, but examples from back reefs, lagoon patches, and reefs near island shores could alternately have been used. All have different thermal environments and should be within the full scope of T monitoring. Stations inside lagoon areas should provide broad geographic distributions, moving from offshore to inshore habitats and, where water depths are sufficient, established at different depths to capture vertical stratification. The arrays forming the transect of the West Channel (Figure 6) are one extreme of such lagoon arrays. The Rock Island areas of Palau [23], a series of basins separated by sills, have instruments from very shallow to the maximum depths in basins that have minimal water exchange. These inner reef arrays have been important in documenting small scale bleaching events in 2007, 2016, and 2018 [23] for which T information would otherwise have not been available.

### 4.3. Will a More Modest Network Suffice?

A network of 100 or more TLs may not be feasible (or necessary) for many reef locations. The Palau network developed gradually, and early results indicated the benefits of expanding the network. A more limited suite of TLs can focus on areas where data are most needed. If knowing outer slope conditions relative to reefs is desirable, depth coverage is more important than geographic coverage. Ideally, the deepest levels of reefs in a given area are instrumented. Once the outer slopes of fringing or barrier reefs are covered, inshore areas are then important to determine whether such areas are thermally distinct and if there is any depth stratification of T.

Atolls provide a simpler system (than Palau) to document, usually having a broad scale lagoon circulation. T regimes might differ on opposite outer slopes of an atoll, and vertical arrays in two areas might be informative. Patch reefs within lagoons are convenient locations to establish vertical arrays from near-surface to maximal lagoon depths. Reef flats would also be important to instrument, as they may have significantly higher T. Channels through the reef rim are also important locations for monitoring, as they are the only connections between ocean and lagoon of sufficient depth to ingress stratified water from offshore.

Lagoon areas with islands, such as those that occur in Palau, are more complex and often tidal currents course through shallow channels advecting water to new areas that may have warmed over shallow bottoms. Fresh or brackish water may enter lagoons from streams, springs or groundwater flows, and are another area where documentation would be important. In special cases, such as caves, caverns, siphons, and other areas where groundwater intrudes, different T conditions are expected to occur and should be documented.

### 4.4. Need for Long-Term Measurements

Short-term in situ T monitoring may not accurately capture broad patterns, particularly with regard to El Niño/La Niña cycles that produce extreme differences [51]. Furthermore, measurement intervals must be short enough to resolve energetic internal waves, although, with sufficient averaging, their effects on a record with long sampling intervals can be reduced. Monitoring networks can take advantage of geography while small oceanic islands can serve as "mooring" equivalents for some global climate considerations. Present-day technical diving capability has expanded the depth range accessible for diver deployment of instruments.

## 5. Conclusions

We have focused on the techniques for developing and maintaining a network of diver-deployed compact, research-quality T loggers for measuring T from a few meters depth on the reef crest to 90 m on the reef slope. This T network targets a wide variety of environments (reef crest, reef slope, reef channel, atolls, lagoons, pinnacles, and headlands), covers an area from 3° N to 8°30′ N impacted by the NEC and NECC, and for periods over 20 years for some stations.

In terms of ocean physics, the network offers a sometimes astounding view of an energetic environment. With sampling over two decades, we have documented large T signals often invisible to SSST from (1) internal waves on time scales of minutes to hours, (2) El Niño on time scales of weeks to years, and (3) decadal-scale trends of +0.2 °C per decade. The latter is a component of variable sea-level rise in the western Pacific, while the other two signals show 14 °C changes over minutes due to internal bores and over weeks during the termination of El Niño and a dramatic blockage of the NECC's usual path. The T network data have been used to create a regression model with SST and SSH capable of predicting depth-varying thermal stress from satellite measurements, which can be tested now at other locations in the tropics. The large temporal, horizontal, and vertical variability noted by the network has further implications for thermal stress on the reef.

In terms of biology, the data points to numerous areas of investigation, although the program was focused initially on obtaining definitive data on the physical environment that could be correlated with events such as coral bleaching. In general, the program has pointed out a dearth of definitive thermal information for most coral reef habitats within Palau and elsewhere, which undermines the ability to interpret biological events from the most basic physical perspective.

**Author Contributions:** P.L.C. conceived and developed the network, fieldwork, data analysis and archiving, and writing and editing. T.M.S.J. Field worked on oceanography, analysis and interpretation of data, and writing and editing. All authors have read and agreed to the published version of the manuscript.

**Funding:** Loggers for the network came from a variety of sources: the University of Oregon, University of California San Diego (UCSD) Senate Marine Sciences Grant, Coastal Observing Research and Development Center of Scripps Institution of Oceanography (UCSD), University the Scholars Fund of UCSD, the David and Lucile Packard Foundation, the University of Delaware School of Marine Science and Policy, the Office of Naval Research FLEAT Initiative, and internal funds of the Coral Reef Research Foundation.

**Acknowledgments:** Fieldwork was aided by Matt Mesubed, Emilio Basilius, Gerda Ucharm, Sharon Patris, Lori J.B. Colin, Steve G. Lindfield, Paul Collins, Travis A. Schramek, Eric J. Terrill, Daniel L. Rudnick, Jennifer A. MacKinnon, Jonathan D. Nash, Mark Moline, and others.

**Conflicts of Interest:** The authors declare no conflict of interest.

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
