# Peer review of "Measuring Temperature in Coral Reef Environments: Experience, Lessons, and Results from Palau"

_jmse, doi:10.3390/jmse8090680_

Round 1
Reviewer 1 Report
The authors have described an in situ temperature-monitoring network in the western Pacific Ocean near the Republic of Palau (3-8°N). The network was started in the Republic of Palau after a 1998 coral bleaching event. The authors have presented briefly benefits of using in situ measurements in studies dealing with thermal regimes on coral reefs. It seems strange to me, that the authors have presented sea temperature measurements from this network mainly as the opposition of satellite derived sea surface temperature measurements. In my view, the range of applications of these two methods of obtaining the water temperature of the ocean is different, so these two methods should be used together in studies on thermal conditions on coral reefs. So, it’s better to present this network as very important component enabling recognition of thermal regimes on coral reefs. The subject presented in the paper is valuable for marine science, however in my opinion the ms should be rewritten.
Specific comments are given below.
Page 1, lines 28-33: Remove the lines 28-33.
Page 1, lines 35-37: Incorrect placing of citing in this sentence “Satellite-derived Sea Surface Temperature (SSST) [1,2] is used by coral reef 35 scientists for global perspectives on temperature (hereafter abbreviated as “T”) patterns/trends and 36 coral bleaching status/predictions [3,4,5]”.
Page 2, lines 47-48: Who examined this. Insert citing.
Page 2, line 67: remove “see”
Page 3, line 111: The subchapter 1.1. shouldn’t be in introduction. It should be placed as a separate chapter: the study area and the description of monitoring network
Page 4, Figure 1: Please improve the quality of map. In figure 1a insert a letter in the rectangular, located under 10°N latitude. Then place this letter in the larger rectangular, which seems to be the zoom of this smaller one. I don’t think that it should be “b” as it is currently. Additionally, any information on geographical names in the Figure 1a could be useful. Place coordinates in Figure 1b.
Page 5, line 161: the colon in the title of the subchapter looks strange.
Page 6, line 187: the colon in the title of the subchapter looks strange. The subchapter should start with the text informing, what will be specified then (this first sentence may finish with the colon).
Page 6, line 214: the title of the subchapter is imprecise
Page 7, line 244: the colon in the title of the subchapter looks strange. The subchapter should start with the text informing, what will be specified then (this first sentence may finish with the colon).
Page 12, line 445: the colon in the title of the subchapter looks strange.
Page 12, lines 451-453: incorrect font style and size
Page 12-18, Results: Highlight your results in the results section. Now all the chapter, with subchapters 3.2-3.6, includes also discussion with results published in cited literature. Separate your results from other research. Then discuss your results in the Discussion section. For example - Page 12, subchapter 3.2. Outer reef temperatures over time and depth of “coral reef conditions”: in this subchapter place an example of your research. In my view, subchapter 3.2. consists mainly of information on work published in [14], [29], [35]. Indicate clearly which results are from your research and which were published in [14], [29], [35].
Page 12, Results: titles of subchapters in this chapter have started to finish with full stop. Why? Unify throughout all the text.
Page 12, lines 455-459: I don’t understand the idea of placing this four-line subchapter. It seems to be an introduction to some chapter/subchapter and should be moved there.
Page 12, lines 461-464: correct grammar in this two sentences. Moreover rather than “Colin 2018” insert the number of reference.
Page 12, line 467: in the ms different symbols for degree are used. Unify using “°”
Page 13, Figure 3: Please improve quality of Figures 3 a and b. Use coordinates and larger size fonts. Currently are unclear. In Figures 3 c and d place units in brackets in titles of vertical axis. In figure 3c in the title of the vertical axis write “weekly mean …”
Page 14, Figure 4: place units in brackets in the title of vertical axis
Page 14, lines 514, 517, 522, 563, 564, 579, and many other examples – use correct symbol for a degree symbol.
Page 15, Figure 5: place units in brackets in the title of vertical axis
Page 15, Figure 6: place units in brackets in titles of vertical axis, explain the meaning of numbers in Fig 6a (the vertical axis – 27-33?, the horizontal axis - 26-30?)
Page 16, Figure 7: place units in brackets in titles of vertical axis, add a title of vertical axis in Figure 7 b
Page 17, Figure 8: place units in brackets in the title of vertical axis,
Page 18, Figure 9: place units in brackets in the title of vertical axis. Is the obtained regression equation statistically significant (level α=0.05)?
Page 18, subchapter 4.1.: In my view, both methods should be used together in studies on thermal conditions on coral reefs. Extend the subchapter and write broader about benefits of using satellite data from both infrared and microwave sensors. Discuss possibilities and limitation of using them and on that backgroud present results from your network. Indicate common applications leading to recognition of thermal conditions on coral reefs. In my opinion, it’s better to present this network as very important component enabling recognition of thermal regimes on coral reefs.
Page 20, the last chapter should be named Conclusions and include conclusions from the research
Page 20, line 681 – erase the word “mainly”
Reviewer 2 Report
Comments to the Author
Reviewer recommendation and comments for Manuscript ID: jmse-891868, entitled “Measuring Temperature in Coral Reef Environments: Experience, Lessons and Results from Palau" for Journal of Marine Science and Engineering.
General Comments:
This study provides insights about the usefulness of temperature for the monitoring of coral reefs and bleaching events. Authors have used a long-term in situ temperature data for this study. I would recommend it for publication provided the following comments be addressed.
- Page 1 lines 28 to 33, I think the authors forgot to remove these instructions provided for the preparation of the manuscript. Please remove them.
- Page 2 lines 48 to 55, do not fit in the introduction. I would suggest to move them in the data used section.
- Page 3 line 98, as mentioned by the authors in line 62 that satellite derived temperatures (SSST) are not able to provide temperature estimates below the surface, I wonder how authors were able to find correlations with in situ T at "all depths"?
- Page 4 Figure 1, there are two Figures with title "(b)", please correct it.
- Page 5 section 2.1, it presents very general information regarding the selection process but a reader is interested to know how authors had selected the stations in this study? I am unable to find such information.
Round 2
Reviewer 1 Report
The authors have improved the quality of the ms significantly. The research is valuable and worth publishing.
Reviewer 2 Report
Authors have addressed all of my question in the revised version. Therefore, I would recommend its publication.